# The Necessity of Magnetic Resonance Imaging in Congenital Diaphragmatic Hernia

**DOI:** 10.3390/diagnostics12071733

**Published:** 2022-07-17

**Authors:** Erick George Neștianu, Cristina Guramba Brădeanu, Dragoș Ovidiu Alexandru, Radu Vlădăreanu

**Affiliations:** 1Faculty of Medicine, “Carol Davila” University of Medicine and Pharmacy, 030167 Bucharest, Romania; radu.vladareanu@umfcd.ro; 2Affidea Medical Center, 022238 Bucharest, Romania; cristinabradeanu@yahoo.com; 3Department of Medical Informatics and Bio-Statistics, University of Medicine and Pharmacy of Craiova, 200349 Craiova, Romania; dragos.alexandru@umfcv.ro; 4Department of Obstetrics and Gynecology, Elias University Emergency Hospital, 011461 Bucharest, Romania

**Keywords:** magnetic resonance imaging, congenital diaphragmatic hernia, ultrasound, lung to head ratio, total lung volume

## Abstract

This is a retrospective study investigating the relationship between ultrasound and magnetic resonance imaging (MRI) examinations in congenital diaphragmatic hernia (CDH). CDH is a rare cause of pulmonary hypoplasia that increases the mortality and morbidity of patients. Inclusion criteria were: patients diagnosed with CDH who underwent MRI examination after the second-trimester morphology ultrasound confirmed the presence of CDH. The patients came from three university hospitals in Bucharest, Romania. A total of 22 patients were included in the study after applying the exclusion criteria. By analyzing the total lung volume (TLV) using MRI, and the lung to head ratio (LHR) calculated using MRI and ultrasound, we observed that LHR can severely underestimate the severity of the pulmonary hypoplasia, even showing values close to normal in some cases. This also proves to be statistically relevant if we eliminate certain extreme values. We found significant correlations between the LHR percentage and herniated organs, such as the left and right liver lobes and gallbladder. MRI also provided additional insights, indicating the presence of pericarditis or pleurisy. We wish to underline the necessity of MRI follow-up in all cases of CDH, as the accurate measurement of the TLV is important for future treatment and therapeutic strategy.

## 1. Introduction

Fetal pulmonary hypoplasia is a rare affliction characterized by the incomplete development of the fetal lung, which can lead to respiratory failure in the newborn and pathological development [1]. Pulmonary hypoplasia has grave consequences on respiratory physiology, especially in the normal gas exchange. It limits the available lung parenchyma and leads to a significantly lower number of viable alveoli for normal respiratory functions [2].

Pulmonary hypoplasia often presents as a secondary cause, associated with other intra or extrathoracic pathologies [3]. Although pulmonary hypoplasia can be found as a primary affliction, this is a very unlikely situation.

It is widely accepted that the true incidence of lung hypoplasia cannot be determined without great difficulty. The literature generally shows an incidence of CDH at 1.4/1000 births and 0.9–1.1/1000 live births [4]. Other authors found an incidence of about 10–11/10,000 live births, but the consensus is that lung hypoplasia is generally underdiagnosed [5]. Mortality rates tend to be high, depending on the severity. Perinatal mortality is high, going up to 75% on the first day after birth. A 47% mortality rate has been observed in the first 60 days of life. Table 1.

Lung expansion is greatly influenced by the point at which a limiting factor begins to obstruct normal development. The pseudo-glandular phase of lung development is generally considered to be completed between the fifth and the seventeenth week of gestation. In this stage, the bronchi arborization is defined, and the lung vascularization develops at a rapid rate [6]. Another important factor in lung development in this stage is the mechanical stimulation provided by the respiratory movement of the fetus. These movements start after about 10 weeks of pregnancy and determine the expansion of the pulmonary parenchyma, the stimulation of epithelial cells, and the movement of fluid through the airways [7].

Normal development of the lung is also highly dependent on the quantity of amniotic fluid. A problem that affects thoracic cavity volume, the presence of respiratory movements, or the adequate quantity of amniotic fluid can lead to the development of pulmonary hypoplasia [8]. These are all considered secondary causes of pulmonary hypoplasia [4].

Pulmonary hypoplasia also appears in congenital pulmonary airway malformation (CPAM), previously known as congenital cystic adenomatoid malformation (CCAM). This is a congenital disorder that occurs in approximately 1/30,000 pregnancies [9]. Other statistics show incidences from 1/30,000 to as high as 1/8300 live births [10].

Among the causes of pulmonary hypoplasia just mentioned, the most frequent is CDH. It is worth underlining the importance of knowing the differential diagnosis of CDH that can be seen in the tables above, because the treatment and planning for each one are different and incorrect implementation can cause harm. This study aimed to observe the two main methods of assessing lung volume in patients with congenital diaphragmatic hernia (CDH) and to evaluate their ability to accurately determine the severity of the illness. Understanding the severity of the illness is essential for determining the outcome of the pregnancy, the necessary antenatal interventions for the fetus, and the best and most efficient treatment during the postnatal period.

The two methods used to evaluate fetal lung capacity are ultrasound investigation, which can approximate lung volume by calculating the lung-to-head ratio (LHR), and the more accurate method using fetal magnetic resonance imaging (MRI), to calculate the total lung volume (TLV) and compare it to reference values.

The incidence of CDH as the sole malformation observed in pregnancy is extremely variable: this is probably due to the variety of reporting methods. The Fetal Medical Foundation and other authors evaluate the incidence to be approximately 4/10,000 live births [11]. It is also more common in the male population with a male/female ratio of 1:0.69 [12] A large study conducted by the European Surveillance of Congenital Anomalies (EUROCAT) shows a prevalence of 2.3 per 10,000 births and 1.6 for isolated CDH cases [13].

The most common herniated abdominal organs that ascend into the thoracic cavity are the stomach, intestinal loops, parts of the colon, and the liver. This usually causes a shift of the mediastinum and the heart to the contralateral side. As such, lung hypoplasia can appear bilaterally, not only on the herniated side. Left side herniation is the most common, seen in about 75–90% of cases, followed by right side herniation, in about 10–15% of cases, with bilateral herniation being seen in only about 1–2% of cases [14]. Survival rates vary from 88% for the small and medium-sized hernias, to about 12% in severe cases [15].

## 2. Materials and Methods

This is a retrospective study. We collected data between 2018 and 2021 from three university hospitals in Bucharest, Romania, that specialize in antenatal diagnosis and treatment.

Patients were selected from those who underwent a second-trimester fetal morphology screening in the aforementioned hospitals. It is important to note that these hospitals attract a large number of patients from a wide area. The examined cases therefore included several patients who sought a second opinion investigation and those with complicated cases that lesser clinics were unable to handle. All patients signed a formal consent form before entering the study. Approval from the hospital Ethical Committee was also obtained before the beginning of the study in December 2017.

Patient follow-up was performed as per ISUOG guidelines for the second and third-trimester [16]. For the delivery, patients were guided to specialized centers that could perform rapid and proper postpartum interventions.

Inclusion criteria were the following: naturally pregnant women (not by in vitro fertilization) who had undergone the second-trimester fetal morphology ultrasound in specialized diagnosis centers that had discovered a CDH as the only observable malformation, and who had had a follow-up MRI examination to confirm or complete the ultrasound diagnosis. The ultrasound examination was performed on high-end devices with dedicated software for obstetrics examination.

The MRI examination was performed using a 1.5 Tesla machine, as current guidelines dictate. We used body coils to enhance the image and the following sequences: Fast Imaging Employing Steady State Acquisition (FIESTA, FOW de 450/500 mm, TR of 5.2 ms, TE of 2.4 ms), Single Shot Fast Spin Echo (SSFSE, FOW de 450/500 mm, TR of 534.4 ms, TE of 160.2 ms), Diffusion Weighted Image (DWI, FOW de 450/500 mm, TR of 6.2 ms, TE of 3.1 ms) and Liver Acquisition with Volume Acceleration (LAVA, FOW de 450/500 mm, TR of 6.2 ms, TE of 3.1 ms). The slice thickness was between 4 and 6 mm. The investigation was made with the mother in a supine or lateral position, without the administration of sedation. The patients were rigorously monitored throughout the pregnancy, with follow-up ultrasounds in the second and third trimesters.

Fetal lung volume was calculated by tracing the lung area of each slice and multiplying it by the thickness between the slices, whilst also accounting for the overlap of the slices. We used the usual DICOM viewing software such as RadiAnt DICOM Viewer Figure 1.

We also used a freeware program to obtain a 3D rendering of the lungs so that the modified anatomy might be better-visualized [17]. Figure 2.

ITK-SNAP is a software application used to segment structures in 3D medical images and is a general-purpose interactive tool for image visualization, manual segmentation, and semi-automatic segmentation Figure 3.

We analyzed the LHR as well as the total lung volume. The LHR was calculated using the area tracing method proposed by Jani et al. [18] and was compared to the LHR obtained via ultrasound to see if there was a discordance between the two methods [19]. For better analysis between our observed LHR values and those expected from the literature, [20] we calculated the percentage represented by our LHR values as compared to the expected values [21]. We called this new parameter the lung to head ratio percentage (LHRP).

We measured the TLV and, similarly to the LHRP, we calculated the TLV percentage (TLVP) in our cases by comparing it to reference values [22]. The TLVP was compared with the LHRP, and we searched for a correlation between the two [18,19].

Data were recorded using Microsoft Excel files; statistical analysis was performed using MS Excel (Microsoft Corp., Redmond, WA, USA), together with the XLSTAT add-on for MS Excel (Addinsoft SARL, Paris, France). Descriptive analysis of the study groups was performed with MS Excel. Statistical tests (Mann–Whitney test for the comparison of non-parametric numerical data, Pearson’s correlation test) were performed using the XLSTAT add-on.

## 3. Results

From a total of 29 patients that we diagnosed with CDH, 22 were enrolled in the study. The others were excluded either because they had other associated anomalies or because they were conceived through in vitro fertilization.

The LHR showed no significant variation between the values obtained through MRI and ultrasound, as we observed a maximum difference of 5%. Although ultrasound examination is an operator-dependent measurement, given that the ultrasounds were not always performed by the same doctor, we concluded that there was no significant variation between the two procedures. Furthermore, the differences did not change the prognosis of the patient.

The 3D rendition of fetal lungs, surrounding organs, and large blood vessels allows for a better understanding of the whole frame of the CDH and for the rest of the herniated organs to be taken into account in preparation for postnatal surgery. This can be done after discussing the case with the surgeon to see if they consider it to be beneficial for the management of the case.

MRI better described and identified the herniated structures in the thoracic cavity and provided useful insight in some cases where ultrasound was not able to accurately describe the pathology. This was particularly useful in differentiating the collapsed lung parenchyma from the herniated liver parenchyma that ascended through the diaphragmatic defect. In some cases, ultrasound cannot accurately differentiate between the two structures, thus also impeding the correct calculation of the LHR. It also showed the herniation of some other organs that might ascend in the thoracic cavity, such as the kidneys and the spleen. Both of these are structures that are often difficult to accurately analyze using ultrasound in cases of CDH. Finally, the MRI examination also showed associated problems, such as pericarditis and pleurisy, that are very difficult to spot using the conventional ultrasound, especially if they are present in small quantities. In one case, the MRI also showed associated mesenteric malrotation, which led to the exclusion of the patient from this study. Figure 4.

Statistical analysis showed a significant correlation between LHRP and some organ herniation, such as that of the left and right liver lobes and the gallbladder, with a p-value lower than 0.05 using the Mann–Whitney method. No other statistically significant correlation has been found, which is unsurprising taking into account the small sample size of patients that we had to work with Table 2.

No correlation has been seen between the LHRP and the age of the mother or the herniation side. Regarding the TLVP, no statistically significant correlation has been found regarding the herniated organs. The same applies to the herniation site and maternal age, which is again to be expected given the small number of patients Table 3.

Regarding the correlation between the LHRP and the TLVP, initially, no statistical correlation was found between them, with a p value calculated using the Pearson correlation coefficient of 0.074524 as seen in the correlation graph Figure 5.

However, we then observed that if the three (13%) most extreme values were removed, the new p value was significantly reduced, to 0.007447, and gained statistical significance. Figure 6.

It is interesting to note that two of the three cases with the most extreme values (meaning a difference between the LHRP and the TLVP bigger than 45%), came from patients presenting a right-sided CDH; only one of the three presented a left-sided herniation.

In two cases (9%), although both the LHR and the pulmonary volume showed lower than normal values, the LHR actually overestimated the volume loss to as much as 9.44%.

In one patient (4.5%), the observed LHR value was slightly above the expected LHR value, even though the patient presented a volume loss of approximately 56.8% according to the MRI examination. In this case, the herniation defect was seen on the left side, showing that in some cases, even if the clinician can observe a CDH, the LHR value might be of no help in quantifying it.

Nine (40.9%) patients had an LHR value between 1 and 1.4; whilst the rest (59.1%) had an LHR value greater than 1.4; which classifies them as having the moderate and respectively mild severity forms, according to the literature, [19]. However, looking at the TLV loss, all 22 cases in our study had more than a 25% volume loss and, as such, are likely to have a poor prognosis according to the literature [23].

Due to the small sample of patients, further research is needed to confirm our findings and to see if new correlations might appear regarding the other parameters studied. The fact that CDH is a rare condition with few specialized centers in Romania able to treat it makes this study a slow process, but we hope that in the future our patient numbers can increase to a more impressive size.

## 4. Discussion

Although CDH occurs only rarely, the unfavorable prognosis of newborns in this situation is enough to justify the necessity of accurate and early diagnosis, along with well-informed family counseling. After the second trimester ultrasound diagnosing the CDH, the patients were followed according to the current guidelines [24,25]. The majority developed without intrauterine intervention. Fetoscopic endotracheal occlusion was deemed necessary in three cases, but only after the MRI examination, as the ultrasound alone was not considered accurate enough to warrant this intervention.

Although most of the mothers were under the age of 35, no statistical correlation was observed between the age of the mother and the likelihood of CDH, nor did we find any literature supporting this theory [26]. It is usually expected that a diagnosis of CDH will be made before 22 weeks of pregnancy, corresponding to the second trimester ultrasound as suggested in the current guidelines. This was also the case in our study. MRI follow-up was performed in the third trimester for 10 cases and in the second trimester for 12 cases, with gestational age ranging between 21.3 to 34.2 weeks.

The majority of patients, 13 out of 22 total patients, were male (59%). The most frequent site of herniation was on the left side, with 17 total cases (77.2%), findings that are in concordance with the published literature. In 16 cases (72.7%), we encountered an anterior herniation site, with only 8 (36.3%) cases having a posterior herniation site. In concordance with the abovementioned findings, in 17 (77.2%) patients we encountered a left hypoplastic lung, corresponding to the left herniation point, and only 5 (22.7%) patients with right lung hypoplasia. These changes also brought a mediastinal shifting corresponding to the herniated location; thus in the cases with a left herniation location, the mediastinum was pushed to the right and vice versa.

We observed no significant difference between the ultrasound LHR value and the one obtained using MRI; both were calculated using the area tracing method proposed by Jani et al. [18]. It is worth noting that, although all patients presented with a lower TLV than initially expected according to the volumetric MRI measurements, the LHR was generally not an accurate measure of the grade of pulmonary volume loss, in most cases underestimating this value. This was seen in 20 patients (9.9%). The greatest discrepancies were seen in three patients (13.6%) where a difference greater than 45% was observed between the LHRP and the TLVP. Two of these were patients with the CDH localized on the right side. This suggests that the LHR measurement might more frequently and more severely underestimate the true magnitude of the problem in patients with a right-sided herniation point. Further study is required to make a final statement on this matter as no significant correlation was underlined in our study.

Data analysis shows a discrepancy between the observed LHR and the observed TLV calculated using MRI. Although some cases presented with an acceptable LHR value, which is often presented as having a good prognosis, we found that these patient’s total lung volume might be as low as 40% of the expected volume and that lower LHR values can indicate a total lung volume of as low as 20% of the expected volume. The correlation between the LHRP and the TLVP suggests that in medical practice the severity of some cases may be greatly underestimated if the physician takes into account only the LHR value, increasing the mortality and morbidity of the disease. We argue that for better management of the CDH after the initial ultrasound diagnosis an MRI should become the new standard for evaluating the status of the lungs, any herniated organs, and the diaphragmatic defect. As previously mentioned, using MRI can lead to new pathological discoveries, such intestinal malrotation, or other malformations that could have evaded detection by ultrasound.

The 3D segmentation of the fetal lung also begins to show its value, as underlined in recent studies such as Davidson et al. [27] and Uss et al. [28]. It is well known that fetal movements during MRI scans can greatly reduce the image quality and make accurate volume measurements a great challenge. We therefore find the deformable slice-to-volume registration method a great opportunity for improving pulmonary volume calculation. It also comes with the advantage that even very motion-corrupted scans can offer correct information, thus saving valuable time in deciding the next medical procedures without the need to reschedule the scan at a future date. This new acquisition method also offers the possibility of realizing multiplanar reconstructions of the region of interest. Although the 3D segmentation program we used lacks the finesse of these newer and more advanced methods, it can be useful in certain cases and can greatly aid the surgeon in planning an intervention.

It has been recently proven that good and timely surgical intervention can result in an important lung volume recovery, Adaikalam et al. [29]. Although our study aims to find a better way to stage and analyze the fetal lung, such studies that prove the possible recovery of the lung parenchyma underline the importance of good and standardized treatment protocols. Adaikalan et al. demonstrated consistent pulmonary volume recovery in infants that had had no previous antenatal treatment. We believe that it would be interesting to see the results of a similar study centered on patients who also underwent antenatal treatments, such as fetoscopic tracheal occlusion. If we look at other materials from recent years, we can see that fetoscopic tracheal occlusion alone is proven to increase survival rates and prognoses in severe cases [30]. Some more recent studies even argue that early fetoscopic tracheal occlusion might bring further benefits to the patient [31]. It is important to focus on these findings, as the guidelines that dictate the most beneficial moment for antenatal procedures might change, as well as those for the patient selection criteria.

The discovery of a discrepancy in the severity gradation of the disease using the LHR value as opposed to directly calculating the TLV using MRI is very important, as several patients might have otherwise been considered lower risk and so not have benefited from correct treatment and follow-up. As more studies emerge, it is important to create good procedures and protocols so that patients receive the correct treatment at the right time. As seen above, timing is extremely important in treating CDH, as many factors influence lung development, especially in the early stages of pregnancy. Delayed treatment might prevent patients from making a satisfying recovery, increasing morbidity and mortality.

## 5. Conclusions

CDH can range from small, asymptomatic defects with little volume loss, to large breaches of the diaphragm causing a vital thoracic volume loss, augmenting the risk of fetal death and respiratory failure after birth. Prenatal sonography has been used for routine screening in pregnancy for many years. This is the main method of prenatal surveillance and also the primary tool for the widespread detection of CDH. Regarding the accuracy of the morphologic description given by the ultrasound in comparison with the MRI, we have observed that ultrasound is effective in identifying most herniated organs but sometimes struggles to offer a comprehensive description of the aberrant morphology. Ultrasound has some more limiting factors: it has trouble differentiating between the intestinal loops and the colon, and it may also lead to difficulty in differentiating between the collapsed lung parenchyma and the liver parenchyma. MRI shows a greater capacity for identifying pericarditis and the ascension of other less common herniated organs such as the kidney and the spleen into the thoracic cavity (although this is a rare occurrence). Clearly, MRI offers a more accurate description of the pathology, although we must note that fetal movements can greatly hinder the examination, sometimes necessitating a repetition of the acquisition process multiple times.

As the the LHR value and the MRI volume measurement can show quite large differences in estimating the severity of the lung hypoplasia, we believe that all patients diagnosed with CDH should undergo a follow-up MRI to more accurately assess the severity of their issues and potentially determine further interventions. The necessity of a good follow-up and family counseling program in cases of CDH cannot be understated; some cases of severe lung hypoplasia might be eligible for late-term pregnancy interruption, bringing great distress to the family. Although it is a rare disease, the morbidity and mortality that come with CDH are enough to warrant thorough methods of diagnosis and treatment. These interventions have a positive impact on the well-being and life expectancy of the patients, reducing the burden on the medical system that they would otherwise incur. An experienced medical team must be employed to ensure that the treatment of the patients, including any postpartum surgical intervention, can take place without delay and in optimal conditions.

## Figures and Tables

**Figure 1 diagnostics-12-01733-f001:**
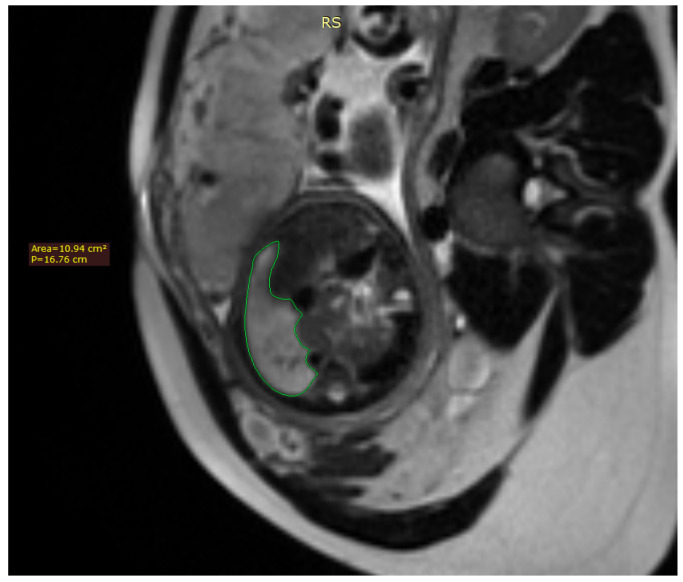
Axial image of the fetus showing the tracing method for calculating the lung volume, using the RadiAnt DICOM Viewer program.

**Figure 2 diagnostics-12-01733-f002:**
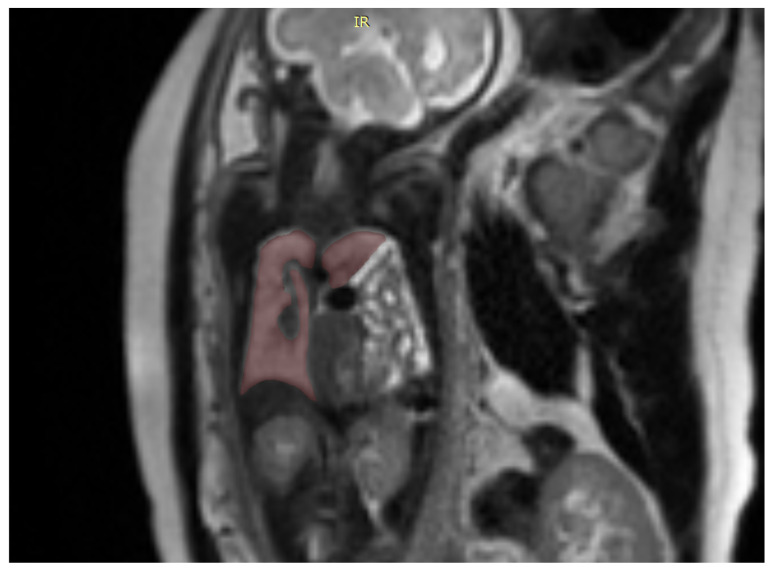
Coronal image of the fetus showing the tracing method for calculating the lung volume, using the ITK-SNAP program.

**Figure 3 diagnostics-12-01733-f003:**
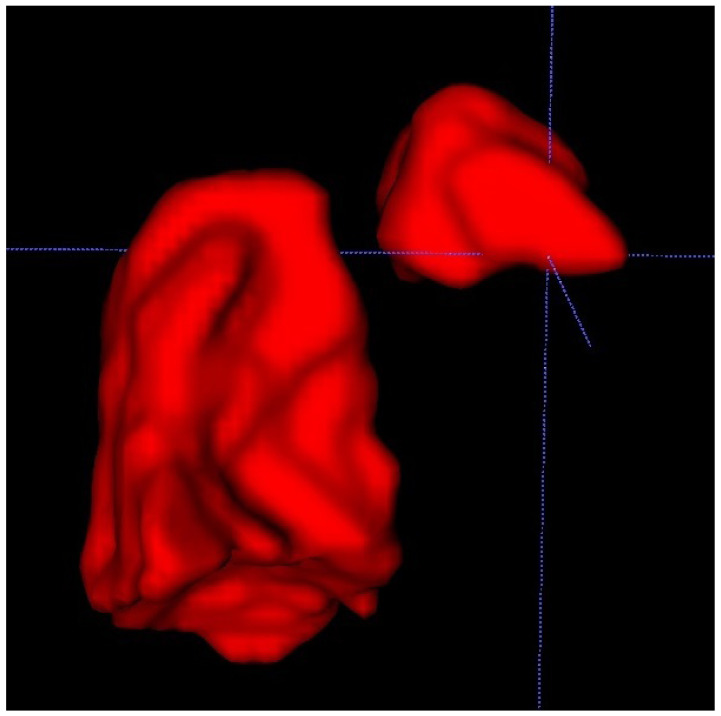
Showing a 3D rendition of the fetal lungs in Figure 1 and Figure 2 using the ITK-SNAP software.

**Figure 4 diagnostics-12-01733-f004:**
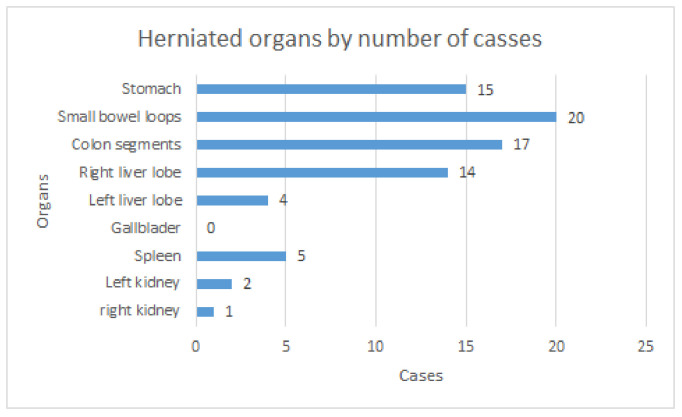
Bar diagram showing the number of cases a specific organ has been identified in the thoracic cavity.

**Figure 5 diagnostics-12-01733-f005:**
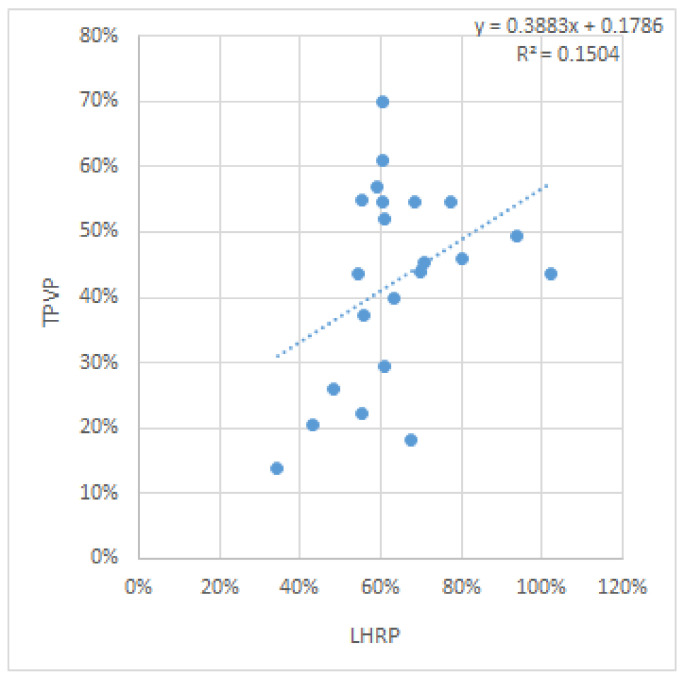
Correlation graphic showing the correlation between the LHRP and TLVP.

**Figure 6 diagnostics-12-01733-f006:**
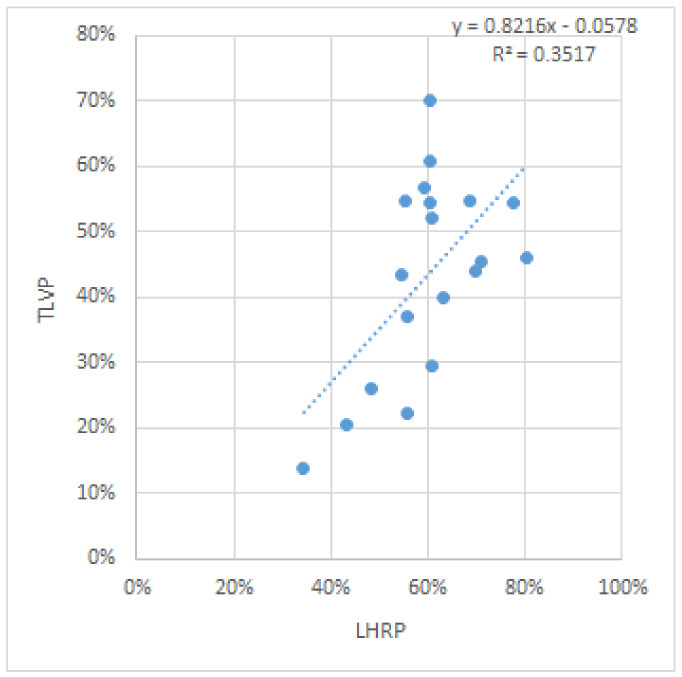
Correlation graphic showing the correlation between the LHRP and TLVP after removal of the three most extreme values.

**Table 1 diagnostics-12-01733-t001:** Causes of intra and extra thoracic cavity volume loss.

Intrathoracic	Extrathoracic
CDH	Oligoamnios—Preterm premature rupture of membranes (PPROM)
Extra lobar sequestration	Skeletal dysplasia
Agenesis of the diaphragm	Large intra-abdominal mass
Mediastinal mass—tumors (mediastinal teratoma)	Neuromuscular condition interfering with fetal breathing
Decreased pulmonary arterial perfusion-cardiovascular anomaly (tetralogy of Fallot) or unilateral absence of the pulmonary artery	

**Table 2 diagnostics-12-01733-t002:** Statistical analysis between the LHRP and various elements.

LHRP	Mean	SD	Mean	SD	p Mann-Whitney
	Age < 35		Age = 35		p
Maternal age	0.67	0.17	0.56	0.07	0.113
	Left		Right		p
Herniation Side	0.61	0.13	0.72	0.21	0.183
	Anterior		Posterior		p
Herniation Site	0.61	0.10	0.65	0.18	0.657
	Left		Right		p
Heart Position	0.60	0.08	0.74	0.24	0.060
Colapsed Lung Side	0.61	0.13	0.72	0.21	0.183
	NO		YES		p
Pleurezy	0.57	0.15	0.67	0.14	0.183
Pericarditis	0.64	0.15	0.62	0.18	0.875
Stomach	0.59	0.10	0.73	0.20	0.091
Intestinal Loops	0.62	0.13	0.85	0.24	0.098
Colon	0.61	0.13	0.73	0.19	0.183
Left Liver	0.56	0.10	0.77	0.15	0.002
Right Liver	0.60	0.13	0.80	0.16	0.024
Gallbladder	0.61	0.13	0.83	0.18	0.035
Spleen	0.65	0.17	0.59	0.04	0.210
Left Kidney	0.64	0.16	0.62	0.02	1.000
Right Kidney	0.64	0.16	0.69	0.00	0.528
Ascitis	0.63	0.15	0.80	0.00	0.207

**Table 3 diagnostics-12-01733-t003:** Statistical analysis between the TLVP and various elements.

TLVP	Mean	SD	Mean	SD	p Mann-Whitney
	Age < 35		Age = 35		p
Maternal age	0.45	0.15	0.37	0.17	0.417
	Left		Right		p
Herniation Side	0.44	0.15	0.37	0.16	0.389
	Anterior		Posterior		p
Herniation Site	0.47	0.14	0.40	0.16	0.473
	Left		Right		p
Heart Position	0.44	0.14	0.38	0.17	0.439
Colapsed Lung Side	0.44	0.15	0.37	0.16	0.389
	NO		YES		p
Pleurezy	0.35	0.17	0.47	0.13	0.152
Pericarditis	0.47	0.13	0.29	0.15	0.0501
Stomach	0.41	0.15	0.46	0.15	0.622
Intestinal Loops	0.42	0.16	0.49	0.08	0.607
Colon	0.42	0.16	0.46	0.10	0.531
Left Liver	0.40	0.15	0.47	0.15	0.453
Right Liver	0.43	0.15	0.41	0.16	0.831
Gallbladder	0.44	0.15	0.36	0.15	0.389
Spleen	0.40	0.15	0.53	0.12	0.117
Left Kidney	0.42	0.16	0.47	0.10	0.909
Right Kidney	0.42	0.15	0.55	0.00	0.344
Ascitis	0.42	0.16	0.46	0.00	0.875

## Data Availability

The data presented in this study are available on request from the corresponding author.

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
