# Peer review of "The Necessity of Magnetic Resonance Imaging in Congenital Diaphragmatic Hernia"

_diagnostics, 2022, doi:10.3390/diagnostics12071733_

Round 1

Reviewer 1 Report

Your idea to work out the advantages of MRI diagnostic in CDH is perfect. But you should rule out the other lung disease, because your paper is a study in CDH and not an overview about lung lesions.

To make a correlation beetween O/E LHR by ultrasound and MRI and comparing it in each patient with relative fetal lung volume by MRI is very important and new to the readers. Therefore you should focus your paper to these results.

In the discussion you are repeating results but do make no relationship to the literature. In your literature year of publication is missing.

You should point out more in detail, that it is necessary to make a safe graduation of severity ( not gravity) of disease by Ultrasound and MRI for planning of antenatal ( Tracheal occlusion) and postnatal treatment in a specialized center.

After profound improvement your paper may be suitable for publication.

Author Response

Good day.
Thank you very much for your constructive advice.

We have revised the article and am ready to resubmit it whit a great number of improvements.

Firstly, I would like to mention that the part in the introduction that references other lung diseases that can cause lung hypoplasia has been a little bit modified. I would like to keep as much of it as possible, as we find it very important that clinicians cand make a good differential diagnosis in the case of CDH. The differential diagnosis that can be made with CDH also include rare pathology that can escape some clinicians, and by presenting them in this paper we hope to spread awareness about them. Of course, if you consider that we should cut out a bit of it, there is no problem, and we will respond accordingly.

We also find the correlation between O/E LHR by ultrasound and MRI to be the center focus of you article, and we made some changes to better underline this aspect. We cand put even greater focus on this aspect if you consider it necessary.

The discussion and results section has been rearranged so that we don’t repeat ourselves as much. Also, we made references to recent literature.
Finally, we made ample language and phrasing corrections so that the message of the article cand be better understood. The bibliography was also modified and is more coherent, also showing the year of publication and providing an easier manner to retrieve it using the DOI index where available.
I do not really know the procedure we must follow, I believe I must submit the revised article now, but I am not entirely sure how to do that.

We are waiting for your response.
Best regards.

Reviewer 2 Report

The  references are inappropriate.

Some statements i.e line 160  to 167  Should be transferred into their section of discussion  while 209- 218 would belong to the section of results.

Author Response

Good day.
Thank you very much for your constructive advice.

The references were modified and are more coherent, also showing the year of publication and providing an easier manner to retrieve them using the DOI index where available.
The discussion and results section has been rearranged as you suggested.

I do not really know the procedure we must follow; I believe I must submit the revised article now, but I am not entirely sure how to do that.

We are waiting for your response.
Best regards.

Reviewer 3 Report

Thank you for this interesting paper. It is undoubtedly of some relevance since it confirms the diagnostic advantage of MRI as additional means of diagnosing CDH.

Unfortunately, one would like to see some substantial changes in its English writing before reviewing the manuscript further. There are many issues in style and spelling in the first lines of the manuscript already. Some examples are listed below.

There are also many issues in the reference section: The citation style is not coherent, and some references lack the date/year of their publication. In addition, some references are hard to retrieve.

line 3: remove comma

line 5: ...from three university hospitals in Bucharest...

line 18: remove "that is"

line 19: which can lead

line 19: newborn instead of new born

line 23: remove "From a cause point of view". 

line 23: Rewrite the sentence! The two phrases after "either" do not have the same structure.

line 25: This study aimed to compare the two main...

I would suggest to have the manuscript checked for language, spelling, conciseness etc. by a native speaker or, at least, to use a software tool (like Grammarly) to revise the manuscript.

Using Grammarly, the first part of your introduction might transform like this:

Fetal pulmonary hypoplasia is a rare affliction characterized by the incomplete development of the fetal lung, which can lead to respiratory failure of the newborn and a pathologic development. [1] Pulmonary hypoplasia has grave consequences on respiratory physiology. It limits the available lung parenchyma, thus lowering the number of viable alveoli to uphold normal respiratory functions. [2] Pulmonary hypoplasia is usually not primary but the sequel to other intra- or extra-thoracic pathologies. [3] This study surveyed the two main methods of appreciating lung volume in patients with a congenital diaphragmatic hernia (CDH) to determine if they can validly evaluate the severity of the illness. This is vital information for deciding the pregnancy outcome. This information is also a prerequisite for determining if the fetus might need antenatal interventions or specialized support in the postnatal period. The two methods used to evaluate the fetal lung capacity were ultrasound, which approximates the lung volume by calculating the lung to head ratio (LHR), and magnetic resonance imaging (MRI) which calculates the total lung volume (TLV) and compares it to reference values.

I would like to refrain from further corrections of style, grammar and spelling issues.

Before resubmitting I would like to suggest to critically revise the references. Besides putting the exisitng references in a coherent style, I would avoid reviews and case presentations. Furthermore I advice checking the newest original research in this field. For example: Davidson et al., Prenat Diagn. 2022.Davidson

J,UusA,EgloffA,etal.Motioncorrectedfetal bodymagneticresonanceimagingprovidesreliable3Dlungvolumesinnormaland abnormalfetuses.PrenatDiagn. 2022;42(5):628

6Davidson

J,UusA,EgloffA,etal.Motioncorrectedfetal bodymagneticresonanceimagingprovidesreliable3Dlungvolumesinnormaland abnormalfetuses.PrenatDiagn. 2022;42(5):628

6Davidson

J,UusA,EgloffA,etal.Motioncorrectedfetal bodymagneticresonanceimagingprovidesreliable3Dlungvolumesinnormaland abnormalfetuses.PrenatDiagn. 2022;42(5):6286

Author Response

Good day.
Thank you very much for your constructive advice.

We made ample language and phrasing corrections so that the message of the article cand be better understood. The bibliography was also modified and is more coherent, also showing the year of publication and providing an easier manner to retrieve it using the DOI index where available.
Finally, we also made references to recent literature in the discussion section.

I do not really know the procedure we must follow; I believe I must submit the revised article now, but I am not entirely sure how to do that.

We are waiting for your response.
Best regards.

Round 2

Reviewer 1 Report

Thank you for improving the paper in a fine manner.

But the differtial diagnosis of lung hypolasia does not need two tables, one will be enough and a compromise.

Author Response

Good day,

I have removed one of the tables as we agreed.
Thank you for the suggestions.

Reviewer 3 Report

Please provide ethics review protocol code and date.

line 70: what do you mean by "incidence of 2.77%"? The cited refence states a prevalence "of 2.3 and 1.6 per 10 000 births for all cases and isolated cases, respectively." based on the EUROCAT-data.

lines 75-77: please provide reference

line 118: "2" should be written as a wird "two"

line 120: there is a space missing in between "by" and "our"

lines 174-176: you sometimes spell out small numbers while you sometimes do not. I would consistently spell out numbers below 10 unless they have a unit, a %, or decimals.

lines 178/189: decimals are usually seprated by a "." not by a ",". In your manuscript, you use a mixture of both.

Author Response

Good day,
I made all the phrasing modifications you underlined in the text. We have also provided references for lines 75-77 and changed the phrasing for line 70 as you suggested.
Regarding the ethics review protocol code and date, I am not sure what you are asking for. I added the date in the text, but other than that, I don’t understand what else I must mention in the text. We have obtained informed consent from the patients and also made the form available to the journal. This article was made as part of a Ph.D. thesis and as such has ethics approval from the university. Please tell me exactly what I must add to the text, as I don’t fully understand this request.
Thank you